# Histopathological and Haemogram Features Correlate with Prognosis in Rectal Cancer Patients Receiving Neoadjuvant Chemoradiation without Pathological Complete Response

**DOI:** 10.3390/jcm11174947

**Published:** 2022-08-23

**Authors:** Yu-Ming Huang, Hsi-Hsien Hsu, Chien-Kuo Liu, Ching-Kuo Yang, Po-Li Tsai, Tzu-Yin Tang, Shih-Ming Hsu, Yu-Jen Chen

**Affiliations:** 1Department of Radiation Oncology, Taipei Hospital, Ministry of Health and Welfare, New Taipei City 242, Taiwan; 2Department of Medicine, MacKay Medical College, New Taipei City 252, Taiwan; 3Department of Biomedical Imaging and Radiological Sciences, National Yang Ming Chiao Tung University, Taipei 112, Taiwan; 4Division of Colorectal Surgery, MacKay Memorial Hospital, Taipei 104, Taiwan; 5Department of Pathology, MacKay Memorial Hospital, Taipei 104, Taiwan; 6Department of Radiation Oncology, MacKay Memorial Hospital, Taipei 104, Taiwan; 7Department of Medical Research, MacKay Memorial Hospital, Taipei 104, Taiwan; 8Department of Artificial Intelligence and Medical Application, MacKay Junior College of Medicine, Nursing, and Management, New Taipei City 252, Taiwan; 9Department of Medical Research, China Medical University Hospital, Taichung 404, Taiwan

**Keywords:** locally advanced rectal cancer, neoadjuvant chemoradiation therapy, tumour microenvironment, intratumoural and peritumoural lymphocytic response, systemic inflammation

## Abstract

Background: Neoadjuvant chemoradiation therapy (NCRT) followed by surgery is the standard treatment for locally advanced rectal cancer (LARC); approximately 80% of patients do not achieve complete response. Identifying prognostic factors predictive of survival in these patients to guide further management is needed. The intratumoural lymphocytic response (ILR), peritumoural lymphocytic reaction (PLR), neutrophil-to-lymphocyte ratio (NLR), and platelet-to-lymphocyte ratio (PtLR) are correlated with the tumour microenvironment and cancer-related systemic inflammation. This study aimed to explore the ability of the ILR, PLR, NLR, and PtLR to predict survival in LARC patients without a complete response to NCRT. Methods: Sixty-nine patients who underwent NCRT and surgery were retrospectively reviewed. The ILR and PLR were assessed in surgical specimens, and the NLR and PtLR were calculated using pre- and post-NCRT blood count data. The Kaplan–Meier method and Cox regression analyses were performed for survival analysis. Results: A high PLR and high post-NCRT NLR and PtLR were significantly associated with better prognosis. Lymphovascular invasion (LVI), post-NCRT neutrophil count, and lymphocyte count were significant predictors of overall survival. LVI and the PLR were independent predictors of disease-free survival. Conclusions: NCRT-induced local and systemic immune responses are favourable prognostic predictors in LARC patients without complete response to NCRT.

## 1. Introduction

Colorectal cancer (CRC) is the third most common cancer and the second leading cause of cancer-related death worldwide [1]. Rectal cancer accounts for approximately 30% of CRC cases and has a 5-year survival rate of 67% [2]. Locally advanced rectal cancer (LARC) is defined as a tumour invading pericolorectal tissues, adjacent organs, or regional lymph nodes. Neoadjuvant chemoradiation therapy (NCRT) followed by surgery (OP) is the standard treatment for LARC and allows for better resectability, anal sphincter preservation, local tumour control, and survival than OP alone [3,4]. Response to NCRT is correlated with survival in patients with LARC. However, only 18.1% of patients achieve a complete response to NCRT [5]. Identifying prognostic factors predictive of survival for the remaining patients to guide further management remains an unmet medical need.

Microsatellite instability (MSI) is a well-known prognostic predictor of CRC. High MSI (MSI-H) is associated with better prognosis than low MSI or microsatellite stability [6]. The prognostic benefit may result from DNA mismatch repair deficiency, which causes mutations that induce antigen and immunological cell infiltration [7]. However, only 15% of CRCs are MSI-H. The tumour microenvironment (TME) consists of immunologic cells, surrounding blood vessels, fibroblasts, signalling molecules, and the extracellular matrix surrounding the cancer cells, and these factors may contribute to tumour occurrence, development, invasion, and metastasis [8]. The intratumoural lymphocytic response (ILR) and peritumoural lymphocytic reaction (PLR) are important components of the TME and may be related to tumour progression and treatment outcome [9]. The ILR is a measurement of tumour-infiltrating lymphocytes (TILs), which can trigger the lysis of cancer cells. The PLR presents as peritumoural Crohn’s-like lymphoid aggregates. The ILR and PLR are also prognostic predictors of CRC [10,11]. Further, the ILR and PLR have been used for phenotypic characterisation of MSI. Therefore, these histopathological features could be used to interpret the immune phenotype of the TME. Finally, there are long-term follow-up data on patients in whom the ILR and PLR were evaluated, which allows assessment of the correlation between the TME and NCRT effects.

The interaction between the local immune response and systemic inflammation is the seventh hallmark of cancer. This interaction is associated with the initiation, development, and progression of several types of malignancies [12]. White blood cell, neutrophil, lymphocyte, platelet, and C-reactive protein levels are closely related to cancer-related inflammation [13]. Combinations of these parameters, including the neutrophil-to-lymphocyte ratio (NLR) and platelet-to-lymphocyte ratio (PtLR), play a crucial role in predicting tumour progression, recurrence, and metastasis [14,15,16]. Further, several recent studies have identified NLR and PtLR as prognostic predictors in patients with CRC [17,18].

A previous study demonstrated that NCRT-induced DNA damage resulting in immunogenic cell death can potentially prime T-cell responses in LARC [19]. However, the importance of the local immune response and systemic inflammation in patients with LARC receiving NCRT is not fully understood. This study aimed to investigate the significance of NCRT-induced local and systemic immune responses by evaluating prognostic predictors, such as the ILR, PLR, NLR, and PtLR, in LARC patients without a complete response to NCRT.

## 2. Materials and Methods

### 2.1. Patients

We retrospectively reviewed newly diagnosed LARC patients who received NCRT followed by OP at one institution between September 2010 and August 2017. Contrast-enhanced CT or MRI was performed for diagnosis and staging. All included patients had transmural (tumour invasion into pericolorectal tissues, visceral peritoneum, or other organs) or node-positive tumours. According to the 7th edition of the American Joint Committee on Cancer staging system, all patients had at least stage II disease (cT3-4 or N+ disease). All patients had an Eastern Cooperative Oncology Group performance status score of 0–2. Further, all patients were older than 18 years, had biopsy-proven rectal cancer, and underwent NCRT followed by total mesorectal excision as the standard OP procedure. Patients with a history of cancer or metastatic disease were excluded from this study. This study aimed to examine the prognostic power of intratumoural and peritumoural lymphocytic responses in patients with LARC after NCRT. If pathological complete response was observed, the post treatment tumour could not be precisely localised. Therefore, five patients with a pathological complete response were excluded. A total of 69 patients were enrolled in this study (Figure 1).

### 2.2. Treatment Protocol

All patients underwent planning CT in the supine position and were immobilised with an alpha cradle. Planning CT images with a slice thickness of 3 mm were acquired through the entire abdomen. Contrast-enhanced CT was used to localise and assess the enhancement patterns of the lesions. Gross tumour volume (GTV) was delineated using diagnostic and simulation images of the primary tumour and/or lymphadenopathy. The high-risk clinical target volume (CTV-H) was determined by expanding the GTV margin by 5–10 mm to consider areas at a significant risk of microscopic disease. The low-risk CTV (CTV-L) was used to cover the mesorectum and perirectal lymphatics, and the external iliac nodes were included for tumours involving gynaecological or genitourinary organs (cT4) [20]. The planning target volume (PTV) was generated by adding a 5–10 mm margin to the CTV in all directions to account for setup error. Radiotherapy (RT) was delivered using either three-dimensional conformal radiotherapy or intensity-modulated radiation therapy based on physician preference. Treatment plans were designed using 6-MV or 10-MV photons. All the patients were treated with linear accelerators. Dosimetric parameters, such as the dose of the CTV and the dose to normal organs, were evaluated in an RT planning system (Eclipse Treatment Planning System; Varian Medical Systems Inc., Palo Alto, CA, USA). The prescribed doses were 50 Gy for the CTV-H and 45 Gy for the CTV-L. The goals were to deliver the prescribed dose to ≥95% of the PTV and to deliver 95% of the prescribed dose to ≥99% of the PTV.

Three different chemotherapy regimens were concurrently administered with RT in this study: continuous intravenous infusion of 5-fluorouracil (5-FU) and leucovorin (LV); oral capecitabine; and oral tegafur-uracil (UFUR) and LV. OP was performed within 6–8 weeks after the last RT fraction. After OP, adjuvant chemotherapy has generally been recommended.

### 2.3. Clinical Data Collection

Medical records were reviewed, and clinical information, including information on patient age, sex, family history, initial carcinoembryonic antigen (CEA) level, clinical stage, and chemotherapy regimen, was collected.

### 2.4. Pathological Analysis

The tumour samples used in this study were derived from post-NCRT surgical specimens, which were intact for comprehensive analysis. All the tumours were reviewed by two experienced gastrointestinal pathology specialists, who were blinded to patients’ clinical data. Tumour size, tumour grade, lymphovascular invasion (LVI), perineural invasion (PNI), margin status, and pathological stage, including the tumour (ypT) and regional lymph node (ypN) statuses, were recorded for survival analysis. Lymphocytic responses, including the ILR and PLR, were assessed according to the cancer reporting protocol released by the College of American Pathologists in 2013. The ILR was categorised into three levels: none (no lymphocytes), mild to moderate (1 or 2 lymphocytes per 400× high-power field), and marked (3 or more lymphocytes per 400× high-power field). The PLR was similarly graded as none, mild to moderate, or marked. If the lymphocytic response was graded as none, it was considered negative. Otherwise, it was considered positive (Figure 2).

### 2.5. Blood Count Data Analysis

Blood count data were available at two time points, the first day of NCRT and the day just before OP. All the data gatherers were blinded to the pathology results and date of death or recurrences. Pre- and post-NCRT serum levels of haemoglobin, white blood cells, neutrophils, lymphocytes, and platelets were acquired for analysis. The NLR and PtLR were defined as the ratios of neutrophils to lymphocytes and platelets to lymphocytes, respectively. The NLR in healthy populations is between 0.7 and 3.5 [21], and a PtLR higher than 250 is correlated with increased mortality in patients with several cancer types and sepsis [22,23,24]. Therefore, the optimal cut-off values of the NLR and PtLR in this study were 3.5 and 250.

### 2.6. Statistical Analysis

Statistical analysis was performed using SigmaPlot version 12.0 (Systat Software, Inc., Point Richmond, CA, USA). Numerical data are presented as mean ± standard deviation. The primary endpoint was overall survival (OS), and the secondary endpoint was disease-free survival (DFS). Survival curves were plotted using the Kaplan–Meier method, and differences in survival were assessed using the log-rank test. The chi-square test was used to assess categorical variables, and the Mann–Whitney U test was used to assess continuous variables. Univariate Cox regression analysis was performed to determine predictors of OS and DFS. Multivariate Cox regression analysis was used to identify the significant prognostic factors. A two-tailed *p* < 0.05 was considered significant.

### 2.7. Ethical Statement

This study was approved by the Institutional Review Board of MacKay Memorial Hospital with ethics committee approval and informed consent (IRB number: 21MMHIS249e).

## 3. Results

### 3.1. Patient Characteristics

The baseline characteristics of the 69 patients are summarised in Table 1. The median age at diagnosis was 58 years (range: 37–86 years) and 44 (63.8%) patients were men. The median pre-NCRT serum CEA level was 4.9 ng/mL (range: 0.0–413.4 ng/mL). A total of 22 patients were diagnosed with clinical stage II disease, and the remaining 47 were diagnosed with stage III disease. A total of 20 patients were treated with continuous infusion of 5-FU (425 mg/m^2^/day during weeks 1 and 5 of RT) and LV (50 mg/m^2^), 14 were treated with oral capecitabine (825 mg/m^2^ twice daily), and 35 were treated with oral UFUR (200 mg/m^2^ daily) and LV (45 mg/day).

### 3.2. Treatment Outcome

The treatment outcomes are shown in Table 2. The median follow-up time was 52.3 months. Forty-seven (68.1%) patients were alive at the time of the analysis. A total of 17 patients were diagnosed with pathological stage I disease, 28 with stage II disease, and 24 with stage III disease. Thirty-two (46.4%) patients were downstaged after NCRT based on the pathological review. The median OS was not reached, and the median DFS was 36.2 months (95% confidence interval, 12.1–60.3). Local failures were noted in 14 (20.3%) patients, and distant metastases were observed in 32 (46.4%) patients.

### 3.3. Pathological Analysis

Details of the pathological analyses are summarised in Table 3. The mean post-NCRT tumour size was 3.0 ± 2.0 cm. A total of 12 (17.4%) patients had LVI and 24 (34.8%) patients had PNI. All patients had negative margins in the surgical specimen. Twenty-seven patients (39.1%) had ypN+ disease. A total of 21 (30.4%) and 38 (55.1%) patients were ILR- and PLR-positive, respectively.

### 3.4. Blood Count Data Analysis

Pre- and post-NCRT blood count data are presented in Table 4. The median pre- and post-NCRT NLR were 2.6 (range: 1.0–12.9) and 4.7 (range: 1.6–24.2), respectively, and the median pre- and post-NCRT PtLR were 166.0 (range: 72.6–516.3) and 320.9 (range: 107.9–1222.7), respectively.

### 3.5. Survival Analysis

The OS and DFS based on possible predictive factors are presented in Table 5. The LVI and ypN statuses were significantly associated with OS (*p* < 0.001 and *p* = 0.02, respectively) (Figure 3). Additionally, women had a better OS than men, with borderline significance (*p* = 0.05). For DFS, pathological stage, LVI, and ypN status were significant prognostic predictors (*p* = 0.04, *p* < 0.001, and *p* < 0.001, respectively) (Figure 4). Patients with PNI had worse DFS than those without PNI, with borderline significance (*p* = 0.05).

### 3.6. The Correlation of the ILR and PLR with OS and DFS

There were no significant differences in OS and DFS based on the ILR (*p* = 0.97 and *p* = 0.60, respectively). No significant differences were found in OS based on the PLR (*p* = 0.07), but the median DFS was significantly higher among patients with PLR positivity than among those without PLR positivity (*p* = 0.03) (Table 6) (Figure 5).

### 3.7. The Correlation of Pre- and Post-NCRT NLR and PtLR with OS and DFS

The prognostic abilities of the pre- and post-NCRT NLR and PtLR are shown in Table 7. There were no significant differences in OS and DFS based on the pre-NCRT NLR and PtLR (Figure 6). However, the median OS was significantly higher among patients with a high post-NCRT NLR or PtLR than among those with a low post-NCRT NLR or PtLR (*p* = 0.03 and *p* = 0.01, respectively). No significant differences in DFS based on the NLR or PtLR were noted (*p* = 0.78 and *p* = 0.13, respectively) (Figure 7).

### 3.8. Univariate and Multivariate Analysis of Prognostic Predictors Affecting OS and DFS

Univariate and multivariate Cox regression analyses were performed to determine the prognostic predictors of OS and DFS based on the clinical, pathological, and blood count data. Univariate analysis identified age, LVI, ypN, post-NCRT tumour size, neutrophil count, and lymphocyte count as predictors of OS. LVI, post-NCRT neutrophil count, and lymphocyte count were independent predictors in the multivariate analysis (Table 8). Univariate analysis revealed LVI, ypN, and PLR as predictors of DFS. LVI and PLR were independent predictors in the multivariate analysis (Table 9).

## 4. Discussion

In this study, we aimed to evaluate the clinical, pathological, and blood count data of patients with LARC receiving NCRT without a complete response to identify prognostic predictors. In the survival analysis, LVI, ypN, and the post-NCRT NLR and PtLR were significant predictors of OS, and pathological stage, LVI, ypN, and the PLR were significant predictors of DFS. Univariate and multivariate analyses revealed that LVI, the post-NCRT neutrophil count, and the lymphocyte count significantly influenced OS and that LVI and PLR were independent predictors of DFS.

Previous studies have investigated the association between the ILR and PLR and survival in patients with CRC. Rozek et al. examined the clinical and pathological characteristics of survival in 2369 CRC patients and reported that a high ILR and PLR were associated with a statistically significant increase in OS and CRC-specific survival [25]. Ogino et al. also proposed that increased overall lymphocytic reactions in CRC, including the ILR and PLR, were related to a significant improvement in OS and CRC-specific survival, independent of lymphocyte count and MSI phenotype [26]. Väyrynen et al. demonstrated superior survival with high PLR density compared to low PLR density in 567 patients with CRC. Therefore, quantitative evaluation of the PLR is recommended as a prognostic indicator for CRC [27].

There is a relative lack of data concerning the TME as a prognostic predictor for LARC patients receiving NCRT. Chen et al. reviewed a series of 126 LARC patients treated with NCRT and found that the post-NCRT stromal-infiltrating CD8+ T cell density corresponded with tumour regression grades, distant metastasis rates, and DFS [28]. Zhang et al. documented 109 LARC patients treated with NCRT and concluded that high levels of CD4+, CD8+, and PD-L1+ TILs were associated with a favourable response to NCRT, whereas high levels of FOXP3+ TILs were associated with a poor response [29]. Akiyoshi et al. noted that CD8+ T cells may be a key element in the response to NCRT, whereas immune checkpoint molecules could be therapeutic targets to enhance tumour regression [30]. Our data showed that the PLR may be a prognostic predictor of DFS in LARC patients receiving NCRT, which is consistent with the results of previous studies assessing NCRT-induced cytotoxic immune response in the TME of rectal cancer [31].

The NLR and PtLR are indicators of systemic inflammation and are easily affected by factors such as trauma, local or systemic infection, acute coronary syndromes, and malignancies. Some studies have examined the poor pathological response, OS, and DFS associated with an elevated pre- and post-NCRT NLR and PtLR in patients with LARC receiving NCRT [32,33,34,35,36]. However, an increased post-NCRT NLR and PtLR significantly correlated with improved OS in our study. This indicates that NCRT-induced systemic inflammation could be a positive prognostic indicator.

Neutrophils secrete various cytokines to stimulate tumour growth, capillary proliferation, and metastasis. High neutrophil counts can upregulate growth factors such as chemokines to induce tumour progression [37]. Platelets secrete platelet-derived growth factor, vascular endothelial growth factor, platelet factor 4, and transforming growth factor-β to increase angiogenesis, microvascular invasion, and cancer cell extravasation to promote tumour progression and metastasis [38]. Lymphocytes can induce cell death and inhibit cancer cell proliferation and migration. Low lymphocyte counts are associated with a reduced antitumour immune response, which leads to tumour growth and disease progression [39]. Therefore, high neutrophil and platelet counts and low lymphocyte counts are negative prognostic predictors in several malignancies [40,41,42]. In our study, high post-NCRT neutrophil levels were associated with poor OS. However, patients with low post-NCRT lymphocyte counts had better OS than those with high lymphocyte counts, which differed from the results of previous studies [43,44,45]. This implies that decreased lymphocyte counts after NCRT may be associated with a better prognosis.

Few studies have focused on decreases in lymphocyte counts and their impact on response to NCRT in LARC. Wu et al. conducted a prospective phase III study that indicated that decreased lymphocyte counts during NCRT were an independent predictor of tumour regression [46]. Heo et al. assessed the correlation between lymphocyte subpopulation counts during NCRT and tumour responses in LARC and reported that decreased lymphocyte counts during NCRT were related to good treatment responses, especially decreases in natural killer cells [47]. Ishihara et al. verified the effects of radiation-induced apoptosis in peripheral blood lymphocytes, which was associated with the extent of tumour regression in patients with LARC after NCRT [48]. From these studies, we may assume that the intrinsic radiosensitivity of cancer cells and circulating lymphocytes could be correlated, which has been validated by previous studies of RT in various cancer types [49,50,51,52]. The increases of NLR and PtLR after NCRT may be related to systemic inflammation, bone marrow suppression with decreases in lymphocyte counts, or intrinsic radiosensitivity of cancer cells.

In the Kaplan–Meier analysis, we reported that high post-NCRT NLR and PtLR were significantly associated with better OS. Then, we wanted to do a further analysis for the impacts of neutrophil, lymphocyte, and platelet counts separately, not only the ratio. Therefore, the counts not the ratio were used in the following univariate and multivariate Cox regression analyses. In the multivariate analysis, LVI, post-NCRT neutrophil count, and lymphocyte count were independent predictors of OS, and LVI and PLR were independent predictors of DFS. LVI has been considered as a well-known and strong stage-independent prognostic predictor of OS and DFS in LARC. Neutrophils possess an intrinsic ability to secrete cytokines to stimulate tumour growth, so higher post-NCRT neutrophil counts may be related to a poorer OS. In the literature, decreased lymphocyte counts during NCRT have been shown to be an independent predictor of treatment responses. The lower lymphocyte counts may be associated with the better radiosensitivity of cancer cells and circulating lymphocytes and better OS. The PLR is a local not systemic NCRT responder, which may be related to DFS.

Our study had some limitations. First, CD4+, CD8+, PD-L1+, and FOXP3+ TILs were not analysed separately. If additional stains had been done with immunohistochemistry data available, we may have been able to assess the prognostic effects of these individual lymphocyte types. In this study, we found that a high PLR without special stains significantly related to better DFS. This preliminary result may be a practical indicator for clinical use. Second, different lymphocyte subpopulations, i.e., T lymphocytes, B lymphocytes, and natural killer cells were not studied. Different immune functions and treatment responses are observed in each subpopulation; therefore, these subpopulations should be independently inspected. Third, the number of patients in our retrospective study was too small to draw firm conclusions. Fourth, the causal relationship between local immune response, systemic inflammation, and good prognosis remained undetermined and may require further investigation. Fifth, quantitative evaluation of the ILR and PLR density should be performed in future research. Sixth, the ILR and PLR were assessed by two pathologists according to the cancer reporting protocol released by the College of American Pathologists in 2013. The bias of interobserver variability should be examined. Finally, local or systemic inflammation may have influenced our results. All possible aetiologies that could affect the TME or circulating lymphocytes should be surveyed in the future.

## 5. Conclusions

Our preliminary results indicated the prognostic power of LVI, the PLR, and the post-NCRT neutrophil and lymphocyte counts in LARC patients without complete response to NCRT. Therefore, the local immune response and systemic inflammation may play a vital role in LARC. In this study, high post-NCRT NLR and PtLR are significantly associated with better OS, and a high PLR is significantly related to better DFS. This may imply that systemic inflammation induced by NCRT could be connected to the intrinsic immune responses of the patients. If recurrences of metastases occur, the more intrinsic immune responses may be associated with the better responses to salvage treatment and better OS. The PLR, one of the immune phenotypes of the TME, is a local not systemic treatment response indicator of NCRT, and it may be related to DFS. This is the first study to combine clinical, pathological, and blood count data to predict the prognosis of LARC patients without complete response to NCRT using a simple, non-invasive, and non-labour-intensive method in clinical practice. However, considering the small number of cases reviewed, a larger prospective study and more evidence are required in the future. Further clinical investigations and experimental animal models are required to validate our findings.

## Figures and Tables

**Figure 1 jcm-11-04947-f001:**
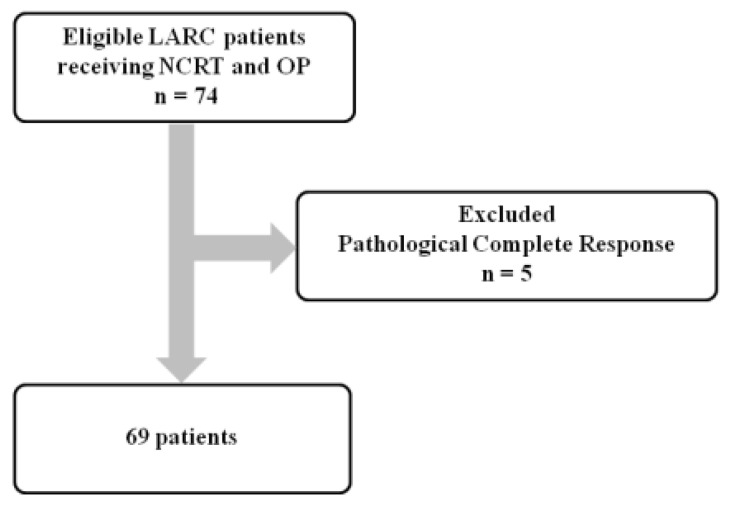
Flowchart of patient inclusion in the study.

**Figure 2 jcm-11-04947-f002:**
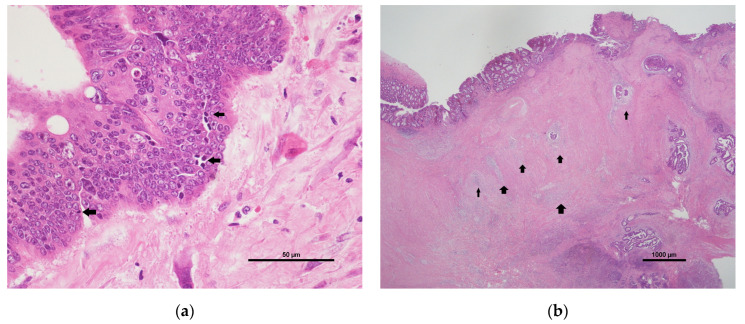
Histopathology of lymphocytic responses. (**a**) The ILR (arrows) indicates lymphocytes present in the neoplastic epithelium (H&E stain, 400×). (**b**) The PLR (arrows) manifests as lymphoid aggregates peripheral to the tumour (H&E stain, 10×). Abbreviations: ILR, intratumoural lymphocytic response; H&E, haematoxylin and eosin; PLR, peritumoural lymphocytic reaction.

**Figure 3 jcm-11-04947-f003:**
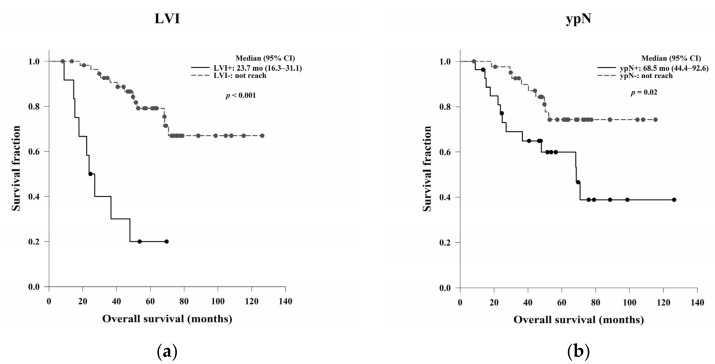
Kaplan–Meier OS curves based on (**a**) LVI and (**b**) ypN status. LVI and ypN positivity were significantly associated with worse OS (*p* < 0.001 and *p* = 0.02, respectively). Abbreviations: LVI, lymphovascular invasion; ypN, pathological lymph node stage after treatment; OS, overall survival.

**Figure 4 jcm-11-04947-f004:**
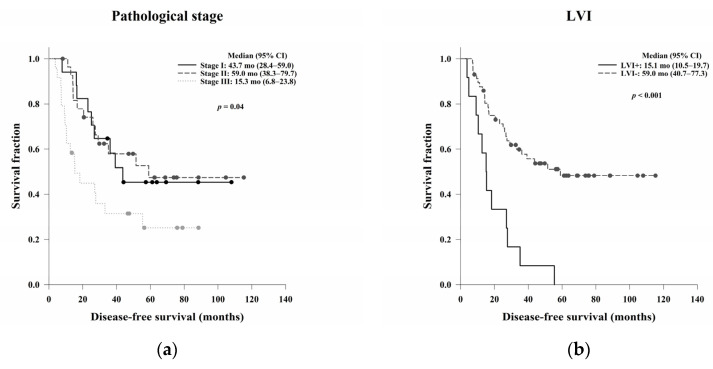
Kaplan–Meier DFS curves based on (**a**) pathological stage, (**b**) LVI, and (**c**) ypN status. A high pathological stage and LVI and ypN positivity were significantly associated with worse DFS (*p* = 0.04, *p* < 0.001, and *p* < 0.001, respectively). Abbreviations: LVI, lymphovascular invasion; ypN, pathological lymph node stage after treatment; DFS, disease-free survival.

**Figure 5 jcm-11-04947-f005:**
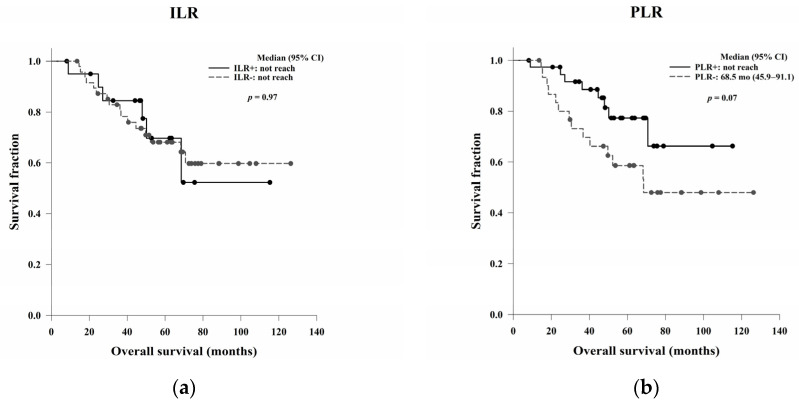
Kaplan–Meier OS curves based on the (**a**) ILR and (**b**) PLR, and Kaplan–Meier DFS curves based on the (**c**) ILR and (**d**) PLR. The ILR and PLR showed no significant prognostic power for OS (*p* = 0.97 and 0.07, respectively). No significant differences in DFS were found based on the ILR (*p* = 0.60), but the median DFS was significantly higher among patients who were PLR-positive than among those who were PLR-negative (*p* = 0.03). Abbreviations: ILR, intratumoural lymphocytic response; PLR, peritumoural lymphocytic reaction; OS, overall survival; DFS, disease-free survival.

**Figure 6 jcm-11-04947-f006:**
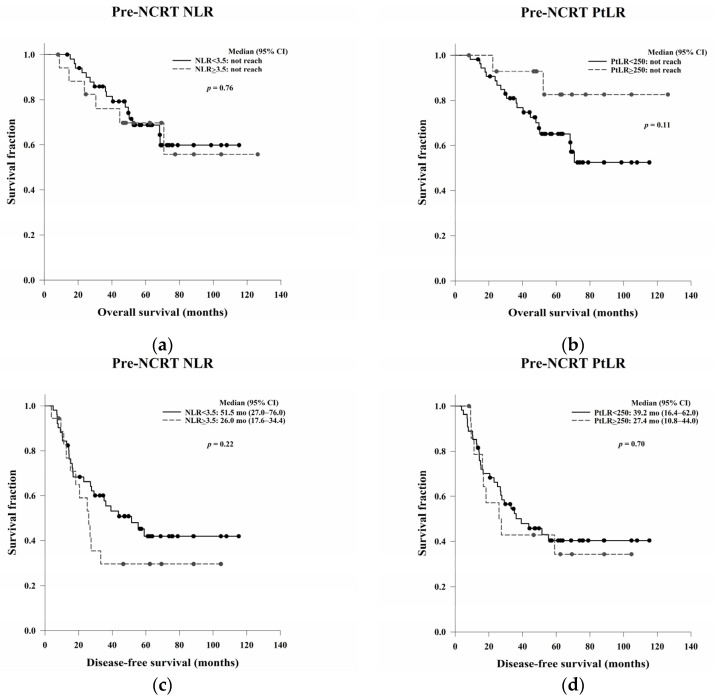
Kaplan–Meier OS curves based on the pre-NCRT (**a**) NLR and (**b**) PtLR, and Kaplan–Meier DFS curves based on the pre-NCRT (**c**) NLR and (**d**) PtLR. The pre-NCRT NLR and PtLR demonstrated no prognostic power for OS (*p* = 0.76 and 0.11, respectively) or DFS (*p* = 0.22 and 0.70, respectively). Abbreviations: NCRT, neoadjuvant chemoradiation therapy; NLR, neutrophil-to-lymphocyte ratio; PtLR, platelet-to-lymphocyte ratio; OS, overall survival; DFS, disease-free survival.

**Figure 7 jcm-11-04947-f007:**
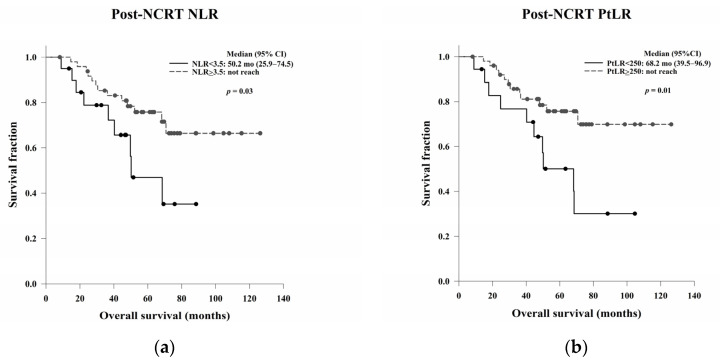
Kaplan–Meier OS curves based on the post-NCRT (**a**) NLR and (**b**) PtLR and Kaplan–Meier DFS curves based on the post-NCRT (**c**) NLR and (**d**) PtLR. The median OS was significantly higher among patients with a high post-NCRT NLR or PtLR than among patients with a low NLR or PtLR (*p* = 0.03 and 0.01, respectively). No significant differences in DFS based on the NLR and PtLR were found (*p* = 0.78 and 0.13, respectively). Abbreviations: NCRT, neoadjuvant chemoradiation therapy; NLR, neutrophil-to-lymphocyte ratio; PtLR, platelet-to-lymphocyte ratio; OS, overall survival; DFS, disease-free survival.

**Table 1 jcm-11-04947-t001:** Baseline characteristics of all patients.

Characteristics	N = 69
Age, median (range), y	58 (37–86)
Sex, N (%)	Male: 44 (63.8)
	Female: 25 (36.2)
Family history, N (%)	Yes: 6 (8.7)
	No: 63 (91.3)
CEA, median (range), ng/mL	4.9 (0.0–413.4)
Clinical stage, N (%)	II: 22 (31.9)
	III: 47 (68.1)
Chemotherapy regimen, N (%)	5-FU + LV: 20 (29.0)
	Capecitabine: 14 (20.3)
	UFUR + LV: 35 (50.7)

Abbreviations: CEA, carcinoembryonic antigen; 5-FU, 5-fluorouracil; LV, leucovorin; UFUR, tegafur-uracil.

**Table 2 jcm-11-04947-t002:** Treatment outcomes of all patients.

Outcomes	N = 69
Pathological stage, N (%)	I: 17 (24.6)
	II: 28 (40.6)
	III: 24 (34.8)
Downstaging, N (%)	32 (46.4)
OS, median (95% CI), m	not reached
DFS, median (95% CI), m	36.2 (12.1–60.3)
DM, N (%)	Yes: 32 (46.4)
	No: 37 (53.6)

Abbreviations: OS, overall survival; CI, confidence interval; DFS, disease-free survival; DM, distant metastasis.

**Table 3 jcm-11-04947-t003:** Pathological analysis of all patients.

Characteristics	N = 69
Tumour size, mean (SD), cm	3.0 (2.0)
Grade, N (%)	I: 13 (18.8)
	II: 47 (68.1)
	III: 9 (13.1)
LVI, N (%)	Yes: 12 (17.4)
	No: 57 (82.6)
PNI, N (%)	Yes: 24 (34.8)
	No: 45 (65.2)
Clear margin, N (%)	69 (100.0)
Pathological T stage (ypT), N (%)	
ypT1	5 (7.2)
ypT2	15 (21.7)
ypT3	48 (69.6)
ypT4	1 (1.5)
Pathological N stage (ypN), N (%)	
ypN−	42 (60.9)
ypN+	27 (39.1)
ILR, N (%)	Yes: 21 (30.4)
	No: 48 (69.6)
PLR, N (%)	Yes: 38 (55.1)
	No: 31 (44.9)

Abbreviations: SD, standard deviation; LVI, lymphovascular invasion; PNI, perineural invasion; ypT, pathological tumour stage after treatment; ypN, pathological lymph node stage after treatment; ILR, intratumoural lymphocytic response; PLR, peritumoural lymphocytic reaction.

**Table 4 jcm-11-04947-t004:** Blood count data analysis of all patients.

Characteristics	Pre-NCRT Levels	Post-NCRT Levels
Hb, mean (SD), g/dL	11.8 (1.9)	11.9 (1.6)
WBC, mean (SD), 10^3^/µL	6976.8 (2398.7)	5252.2 (1720.4)
Neu, mean (SD), 10^3^/µL	4539.5 (2082.1)	3774.3 (1535.7)
Lym, mean (SD), 10^3^/µL	1652.3 (686.3)	774.3 (350.4)
PLT, mean (SD), 10^3^/µL	277,608.7 (81,075.4)	238,695.7 (78,283.3)
NLR, median (range)	2.6 (1.0–12.9)	4.7 (1.6–24.2)
PtLR, median (range)	166.0 (72.6–516.3)	320.9 (107.9–1222.7)

Abbreviations: NCRT, neoadjuvant chemoradiation therapy; Hb, haemoglobin; SD, standard deviation; WBC, white blood cell; Neu, neutrophil; Lym, lymphocyte; PLT, platelet; NLR, neutrophil-to-lymphocyte ratio; PtLR, platelet-to-lymphocyte ratio.

**Table 5 jcm-11-04947-t005:** Comparison of OS and DFS based on clinicopathological factors.

Factors	Mean OS (SD), m	*p* ^1^	Mean DFS (SD), m	*p* ^1^
Age, years		0.31		0.51
<60 (N = 38)	95.7 (8.1)		54.1 (7.7)	
≥60 (N = 31)	73.5 (6.9)		59.1 (8.0)	
Sex		0.05		0.43
Male (N = 44)	73.5 (6.0)		59.3 (6.9)	
Female (N = 25)	107.2 (8.7)		52.0 (9.3)	
CEA		0.06		0.60
<5 ng/mL (N = 36)	102.7 (7.3)		59.5 (7.6)	
≥5 ng/mL (N = 33)	72.3 (7.8)		54.6 (8.2)	
Clinical stage		0.15		0.09
II (N = 22)	96.1 (7.4)		74.4 (10.5)	
III (N = 47)	83.3 (7.9)		47.7 (6.0)	
Pathological stage		0.12		0.04
I (N = 17)	84.2 (8.6)		63.4 (10.2)	
II (N = 28)	95.0 (7.2)		69.1 (9.1)	
III (N = 24)	75.5 (10.7)		35.3 (6.9)	
Chemotherapy		0.49		0.43
5-FU + LV (N = 20)	70.8 (8.8)		44.8 (9.4)	
Capecitabine (N = 14)	68.9 (5.3)		49.4 (7.1)	
UFUR + LV (N = 35)	92.5 (8.2)		63.2 (8.4)	
Grade		0.13		0.78
I (N = 13)	78.9 (11.2)		57.3 (12.9)	
II (N =47)	95.4 (7.1)		59.9 (7.1)	
III (N =9)	50.1 (11.5)		40.9 (11.3)	
LVI		<0.001		<0.001
Positive (N = 12)	33.6 (6.2)		19.6 (4.3)	
Negative (N = 57)	101.1 (6.0)		68.0 (6.5)	
PNI		0.14		0.05
Positive (N = 24)	68.6 (8.4)		38.8 (7.1)	
Negative (N = 45)	97.0 (7.1)		68.4 (7.4)	
Pathological T stage		0.62		0.93
ypT1-2 (N = 20)	95.7 (10.3)		55.4 (9.7)	
ypT3-4 (N = 49)	82.8 (6.5)		59.2 (7.0)	
Pathological N stage		0.02		<0.001
ypN− (N = 42)	96.0 (5.6)		71.3 (7.4)	
ypN+ (N = 27)	73.5 (9.8)		34.5 (6.3)	

^1^ A two-sided *p* value < 0.05 was considered statistically significant. Abbreviations: OS, overall survival; DFS, disease-free survival; SD, standard deviation; CEA, carcinoembryonic antigen; 5-FU, 5-fluorouracil; LV, leucovorin; UFUR, tegafur-uracil; LVI, lymphovascular invasion; PNI, perineural invasion; ypT, pathological tumour stage after treatment; ypN, pathological lymph node stage after treatment.

**Table 6 jcm-11-04947-t006:** Comparison of OS and DFS based on the ILR and PLR.

Factors	Mean OS (SD), m	*p* ^1^	Mean DFS (SD), m	*p* ^1^
ILR		0.97		0.60
Positive (N = 21)	82.6 (10.6)		60.8 (10.5)	
Negative (N = 48)	91.4 (6.9)		54.5 (6.5)	
PLR		0.07		0.03
Positive (N = 38)	92.4 (7.2)		70.9 (7.9)	
Negative (N = 31)	80.2 (8.9)		43.4 (7.3)	

^1^ A two-sided *p* value < 0.05 was considered statistically significant. Abbreviations: OS: overall survival; DFS: disease-free survival; ILR: intratumoural lymphocytic response; PLR: peritumoural lymphocytic reaction; SD: standard deviation.

**Table 7 jcm-11-04947-t007:** Comparison of OS and DFS in based on the pre- and post-NCRT NLR and PtLR.

Factors	Mean OS (SD), m	*p* ^1^	Mean DFS (SD), m	*p* ^1^
Pre-NCRT NLR				
<3.5 (N = 51)	86.0 (5.9)	0.76	62.7 (6.8)	0.22
≥3.5 (N = 18)	87.8 (12.5)		44.4 (9.6)	
Pre-NCRT PtLR				
<250 (N = 54)	80.3 (6.0)	0.11	60.2 (6.6)	0.70
≥250 (N = 15)	111.2 (9.8)		50.6 (11.1)	
Post-NCRT NLR				
<3.5 (N = 20)	57.0 (7.1)	0.03	49.0 (8.7)	0.78
≥3.5 (N = 49)	98.3 (6.6)		59.4 (6.6)	
Post-NCRT PtLR				
<250 (N = 18)	61.5 (8.5)	0.01	42.8 (9.9)	0.13
≥250 (N = 51)	100.1 (6.5)		63.3 (6.8)	

^1^ A two-sided *p* value < 0.05 was considered statistically significant. Abbreviations: OS, overall survival; DFS, disease-free survival; NCRT, neoadjuvant chemoradiotherapy; NLR, neutrophil-to-lymphocyte ratio; PtLR, platelet-to-lymphocyte ratio; SD, standard deviation.

**Table 8 jcm-11-04947-t008:** Univariate and multivariate analyses for predictors of OS.

Predictors		Univariate Analysis			Multivariate Analysis		
		HR	95% CI	*p* ^1^	HR	95% CI	*p* ^1^
Age (y)		1.040	1.003–1.078	0.032	1.040	0.994–1.087	0.087
Post-NCRT tumour size (cm)		1.390	1.138–1.698	0.001	1.073	0.848–1.358	0.555
LVI	Negative	1			1		
	Positive	7.864	3.242–19.072	<0.001	11.888	2.944–48.009	0.001
ypN	Negative	1			1		
	Positive	2.730	1.165–6.397	0.021	1.089	0.298–3.982	0.897
ILR	Positive	1					
	Negative	1.018	0.396–2.613	0.971			
PLR	Positive	1					
	Negative	2.191	0.917–5.235	0.077			
Post-NCRT Neu (/µL)		1.000	1.000–1.001	0.033	1.000	1.000–1.001	0.044
Post-NCRT Lym (/µL)		1.002	1.000–1.003	0.001	1.002	1.001–1.003	0.001
Post-NCRT PLT (/µL)		1.000	1.000–1.000	0.390			

^1^ A two-sided *p* value < 0.05 was considered statistically significant. Abbreviations: OS, overall survival; HR, hazard ratio; CI, confidence interval; NCRT, neoadjuvant chemoradiation therapy; LVI, lymphovascular invasion; ypN, pathological lymph node stage after treatment; ILR, intratumoural lymphocytic response; PLR, peritumoural lymphocytic reaction; Neu, neutrophil; Lym, lymphocyte; PLT, platelet.

**Table 9 jcm-11-04947-t009:** Univariate and multivariate analyses for predictors of DFS.

Predictors		Univariate Analysis			Multivariate Analysis		
		HR	95% CI	*p* ^1^	HR	95% CI	*p* ^1^
Age (y)		0.991	0.965–1.018	0.514			
Post-NCRT tumour size (cm)		1.160	0.985–1.365	0.075			
LVI	Negative	1			1		
	Positive	3.914	1.945–7.873	<0.001	2.895	1.157–7.242	0.023
ypN	Negative	1			1		
	Positive	2.526	1.344–4.749	0.004	1.508	0.659–3.454	0.331
ILR	Positive	1					
	Negative	1.208	0.601–2.431	0.596			
PLR	Positive	1					
	Negative	2.006	1.063–3.786	0.032	1.985	1.046–3.764	0.036
Post-NCRT Neu (/µL)		1.000	1.000–1.000	0.066			
Post-NCRT Lym (/µL)		1.001	1.000–1.002	0.079			
Post-NCRT PLT (/µL)		1.000	1.000–1.000	0.588			

^1^ A two-sided *p* value < 0.05 was considered statistically significant. Abbreviations: DFS, disease-free survival; HR, hazard ratio; CI, confidence interval; NCRT, neoadjuvant chemoradiation therapy; LVI, lymphovascular invasion; ypN, pathological lymph node stage after treatment; ILR, intratumoural lymphocytic response; PLR, peritumoural lymphocytic reaction; Neu, neutrophil; Lym, lymphocyte; PLT, platelet.

## Data Availability

Not applicable.

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
