# Peer review of "Histopathological and Haemogram Features Correlate with Prognosis in Rectal Cancer Patients Receiving Neoadjuvant Chemoradiation without Pathological Complete Response"

_jcm, 2022, doi:10.3390/jcm11174947_

Round 1
Reviewer 1 Report
Thank you very much for the opportunity to review the paper dealing with histopathological and hematological features correlating with prognosis in rectal cancer patients receiving neoadjuvant therapy without complete response.
The authors present a series of 69 patients treated between 2010 and 2017 resulting in a long median follow-up of 52.3 months. The presented OS of 68.1% and distant metastasis rate is honest and of interest. The paper is well written an easy to read. The introduction is interesting and focusing to the topic.
I have some comments and questions to the methods, results and discussion:
Was the study proven by the ethic committee?
Why were patients with complete response excluded? Do you see immunoreaction in these patients as well? How many patients had complete response? The numbers should be presented and the exclusion disussed.
L 96 patients. The authors should present their indication to NCRT. In example distance to the mesorectal fascia, height of the tumor, EMVI on MRI… A flowchart of all patients should be presented. How many patients received NCRT (percentage).
L98: Was the histopathology reviewed again or used old clinical data? If so, why did only on pathologist review the samples? The bias of interobserver variability should be discussed.
L195 Treatment outcome: Do you have data regarding the tumor regression grade? It would be interesting to correlate TRG with immunoreaction.
L 264: The figures would be easier to read if you mentioned OS and DFS instead of survival fraction and if you showed the p value in the graph. Figure 5b shows different curves but the p value is 0.11. It would also be better to present confidence intervals.
To me the conclusion drawn from your results is not quite sound.
There is a difference between OS and DFS and univariate and multivariate analysis.
L 247 Figure 4. In the Kaplan Meier analysis, you present ILR and PLR and only DFS is statistically significant different.
L 270 Figure 6: Post-NCRT NLR and PtLR OS is significantly different but not DFS.
You should discuss this difference
L 286: In table 8 and 9 you do not present the ratio but Neu, Lym and PLT count. You should explain why you choose the count and not the ratio and eventually change the presentation. In the multivariate analysis only LVI and Neu are statistically significant for OS and LVI and PLR for DFS. You should discuss the differences in survival analysis and multivariate analysis. The discrepancies might be due to the mentioned limitations and the small sample size.
The topic of immunreaction to NCRT is of great clinical interest. The presentation is clear but the discussion should be revised.
Author Response
Point 1: Was the study proven by the ethic committee?
Response 1: The study was proven by the ethic committee in MacKay Memorial Hospital. The Material and methods (2.7) is added as follows for explanation.
“2.7. Ethical Statement
This study was approved by the Institutional Review Board of MacKay Memorial Hospital with ethics committee approval and informed consent (IRB number: 21MMHIS249e).”
Point 2: Why were patients with complete response excluded? Do you see immunoreaction in these patients as well? How many patients had complete response? The numbers should be presented and the exclusion discussed.
Response 2: Thanks for your great comment. This study aimed to examine the prognostic power of intratumoural and peritumoural lymphocytic responses in patients with LARC after NCRT. If pathological complete response was observed, the post treatment tumour could not be precisely localized. Therfore, five patients with a pathological complete response were excluded. We focus on the LARC patients receiving NCRT and OP without pathological complete response, so the title is named as “Histopathological and Haemogram Features Correlate with Prognosis in Rectal Cancer Patients Receiving Neoadjuvant Chemoradiation without Pathological Complete Response”. The Material and methods (2.1) is revised as follows for explanation.
Previous version:
“Because this study aimed to examine the prognostic power of intratumoural and peritumoural lymphocytic responses in patients with LARC after NCRT, five patients with a pathological complete response were excluded.”
Revised version:
“This study aimed to examine the prognostic power of intratumoural and peritumoural lymphocytic responses in patients with LARC after NCRT. If pathological complete response was observed, the post treatment tumour could not be precisely localized. Therfore, five patients with a pathological complete response were excluded.”
Point 3: L96 patients. The authors should present their indication to NCRT. In example distance to the mesorectal fascia, height of the tumor, EMVI on MRI… A flowchart of all patients should be presented. How many patients received NCRT (percentage).
Response 3: Thanks for your suggestion. All included patients had transmural (tumour invasion into pericolorectal tissues, visceral peritoneum, or other organs) or node-positive tumours. According to the 7th edition of the American Joint Committee on Cancer staging system, all patients had at least stage II disease (cT3-4 or N+ disease). A flowchart all patients is presented in Figure 1. The Material and methods (2.1) is revised as follows.
Previous version:
“Contrast-enhanced CT or MRI was performed for diagnosis and staging. According to the 7th edition of the American Joint Committee on Cancer staging system, all patients had at least stage II disease (cT3-4 or N+ disease).”
“A total of 69 patients were enrolled in this study.”
Revised version:
“Contrast-enhanced CT or MRI was performed for diagnosis and staging. All included patients had transmural (tumour invasion into pericolorectal tissues, visceral peritoneum, or other organs) or node-positive tumours. According to the 7th edition of the American Joint Committee on Cancer staging system, all patients had at least stage II disease (cT3-4 or N+ disease).”
“A total of 69 patients were enrolled in this study (Figure 1).”
Point 4: L98: Was the histopathology reviewed again or used old clinical data? If so, why did only one pathologist review the samples? The bias of interobserver variability should be discussed.
Response 4: The histopathology of biopsy was not reviewed again. We focused on the tumour samples derived from post-NCRT surgical specimens, which were intact for comprehensive analysis. All the tumours were reviewed by an experienced gastrointestinal pathologist, and the report was verified by another pathologist. Therefore, two pathology specialists blindly reviewed all cases. The bias of interobserver variability is a limitation in this study, and it is added in the limitation section. The Material and methods (2.4) and Discussion are revised as follows.
Previous version:
“The tumour samples used in this study were derived from post-NCRT surgical specimens. All the tumours were blindly reviewed by an experienced gastrointestinal pathologist.”
“Fifth, quantitative evaluation of the ILR and PLR density should be performed. Finally, local or systemic inflammation may have influenced our results. All possible aetiologies that could affect the TME or circulating lymphocytes should be surveyed in the future.”
Revised version:
“The tumour samples used in this study were derived from post-NCRT surgical specimens, which were intact for comprehensive analysis. All the tumours were reviewed by two experienced gastrointestinal pathology specialists, who were blinded to patients’ clinical data.”
“Fifth, quantitative evaluation of the ILR and PLR density should be performed in the further study. Sixth, the ILR and PLR were assessed by two pathologists according to the cancer reporting protocol released by the College of American Pathologists in 2013. The bias of interobserver variability should be examined. Finally, local or systemic inflammation may have influenced our results. All possible aetiologies that could affect the TME or circulating lymphocytes should be surveyed in the future.”
Point 5: L195 Treatment outcome: Do you have data regarding the tumor regression grade? It would be interesting to correlate TRG with immunoreaction.
Response 5: Thanks for your great comment. We didn’t have data regarding the tumour regression grade. In this study, we reviewed the post-NCRT tumour size in surgical specimens as a local treatment response indicator. Univariate analysis identified post-NCRT tumour size as a predictor of OS, which was not significant in multivariate analysis. In literature reviews, Chen et al. reviewed a series of 126 LARC patients treated with NCRT and found that the post-NCRT stromal-infiltrating CD8+ T cell density corresponded with tumour regression grades, distant metastasis rates, and DFS. The Discussion is revised as follows.
Previous version:
“Chen et al. reviewed a series of 126 LARC patients treated with NCRT and found that the post-NCRT stromal-infiltrating CD8+ T cell density corresponded with tumour regression, distant metastasis, and DFS.”
Revised version:
“Chen et al. reviewed a series of 126 LARC patients treated with NCRT and found that the post-NCRT stromal-infiltrating CD8+ T cell density corresponded with tumour regression grades, distant metastasis rates, and DFS.”
Point 6: L 264: The figures would be easier to read if you mentioned OS and DFS instead of survival fraction and if you showed the p value in the graph. Figure 5b shows different curves but the p value is 0.11. It would also be better to present confidence intervals.
Response 6: Thanks for your suggestion. We added OS and DFS in the figures with p value shown in the graphs. In Figure 5b, the pre-NCRT PtLR demonstrated no prognostic power for OS (p = 0.11), but different curves were shown in the graphs. It may be related to very few people in pre-NCRT PtLR ≥ 250 group. The confidence intervals were also presented in the revised figures.
Point 7: To me the conclusion drawn from your results is not quite sound.
There is a difference between OS and DFS and univariate and multivariate analysis.
L 247 Figure 4. In the Kaplan Meier analysis, you present ILR and PLR and only DFS is statistically significant different.
L 270 Figure 6: Post-NCRT NLR and PtLR OS is significantly different but not DFS.
You should discuss this difference
Response 7: In this study, high post-NCRT NLR and PtLR are significantly associated with better OS, and a high PLR is significantly related to better DFS. It may imply that systemic inflammation induced by NCRT could be connected to the intrinsic immune responses of the patients. If recurrences of metastases occur, the more intrinsic immune responses may be associated with the better responses to salvage treatment and better OS. The PLR, one of immune phenotypes of the TME, is a local not systemic treatment response indicator of NCRT, and it may be related to DFS. The Conclusion is revised as follows.
Previous version:
“Therefore, the local immune response and systemic inflammation may play a vital role in LARC.”
Revised version:
“Therefore, the local immune response and systemic inflammation may play a vital role in LARC. In this study, high post-NCRT NLR and PtLR are significantly associated with better OS, and a high PLR is significantly related to better DFS. It may imply that systemic inflammation induced by NCRT could be connected to the intrinsic immune responses of the patients. If recurrences of metastases occur, the more intrinsic immune responses may be associated with the better responses to salvage treatment and better OS. The PLR, one of immune phenotypes of the TME, is a local not systemic treatment response indicator of NCRT, and it may be related to DFS.”
Point 8: L 286: In table 8 and 9 you do not present the ratio but Neu, Lym and PLT count. You should explain why you choose the count and not the ratio and eventually change the presentation. In the multivariate analysis only LVI and Neu are statistically significant for OS and LVI and PLR for DFS. You should discuss the differences in survival analysis and multivariate analysis. The discrepancies might be due to the mentioned limitations and the small sample size.
Response 8: Thanks for your great comment. Following descriptions are added in Discussion for explanation.
“In the Kaplan Meier analysis, we reported that high post-NCRT NLR and PtLR were significantly associated with better OS. Then, we wanted to do a further analysis for the impacts of neutrophil, lymphocyte, and platelet counts separately, not only the ratio. Therefore, the counts not the ratio were used in the following univariate and multivariate Cox regression analyses. In the multivariate analysis, LVI, post-NCRT neutrophil count, and lymphocyte count were independent predictors of OS, and LVI and PLR were independent predictors of DFS. LVI has been considered as a well-known and strong stage-independent prognostic predictor of OS and DFS in LARC. Neutrophils possess an intrinsic ability to secrete cytokines to stimulate tumour growth, so higher post-NCRT neutrophil counts may be related to a poorer OS. In literature reviews, decreased lymphocyte counts during NCRT were an independent predictor of treatment responses. The lower lymphocyte counts may be associated with the better radiosensitivity of cancer cells and circulating lymphocytes and better OS. The PLR is a local not systemic NCRT responder, which may be related to DFS.”

Reviewer 2 Report
The premise is interesting here regarding systemic and local inflammation after chemo-radiation, which could correlate with prognosis. Some consideration should be given to use of immunotherapy in the future, might want to discuss that. There are also some limitations researchers may want to emphasize from the study as well.
1. Were the pathologists blinded to clinical data? There seems some subjectivity to assessment of ILR and PLR. Please describe in 2.4
2. How come additional stains were not done, perhaps to characterize the lymphocytes? Or quantify the infiltration?
3. How were CBC/diff chosen? Wouldn't patients have multiple CBC/diff post NCRT? Are the data gatherers blinded to the pathology results? or date of death/recurrences? Please describe in section 2.5
4. Did any patients receive post-surgery adjuvant chemotherapy? Wouldn't that influence DFS and OS?
5. How do you know that the PtLR and NLR truly related to systemic inflammation/post-treatment reaction rather than bone marrow suppression from chemo-RT? Some people can develop thrombocytopenia or lymphopenia after pelvic radiation for a while.
6. DFS is difficult to interpret because disease recurrence could be local or metastatic recurrence. Could you elaborate on site of recurrence (local vs. distant) and how that could be affected from your 4 prognosis factors?
Author Response
Point 1: Were the pathologists blinded to clinical data? There seems some subjectivity to assessment of ILR and PLR. Please describe in 2.4
Response 1: Thanks for your great comment. The pathologists were blinded to clinical data. The Material and methods (2.4) is revised as follows for explanation.
Previous version:
“The tumour samples used in this study were derived from post-NCRT surgical specimens. All the tumours were blindly reviewed by an experienced gastrointestinal pathologist.”
Revised version:
“The tumour samples used in this study were derived from post-NCRT surgical specimens, which were intact for comprehensive analysis. All the tumours were reviewed by two experienced gastrointestinal pathology specialists, who were blinded to patients’ clinical data.”
Point 2: How come additional stains were not done, perhaps to characterize the lymphocytes? Or quantify the infiltration?
Response 2: This retrospective study was intended to exam the prognostic power of ILR and PLR, and concluded that a high PLR, not high ILR, was significantly associated with better prognosis. Additional stains and quantification of the infiltration were not done. It is a limitation, and a prospective study to characterize the lymphocytes or quantify the infiltration may be needed. In this study, we found that a high PLR without special stains significantly related to better DFS. This preliminary result may be a practical indicator for clinical use. The Discussion is revised as follows for explanation.
Previous version:
“Our study had some limitations. First, CD4+, CD8+, PD-L1+, and FOXP3+ TILs were not analysed separately. If immunohistochemistry data had been available, we may have been able to assess the prognostic effects of these individual lymphocyte types.”
“Fifth, quantitative evaluation of the ILR and PLR density should be performed.”
Revised version:
“Our study had some limitations. First, CD4+, CD8+, PD-L1+, and FOXP3+ TILs were not analysed separately. If additional stains had been done with immunohistochemistry data available, we may have been able to assess the prognostic effects of these individual lymphocyte types. In this study, we found that a high PLR without special stains significantly related to better DFS. This preliminary result may be a practical indicator for clinical use.”
“Fifth, quantitative evaluation of the ILR and PLR density should be performed in the further study.”
Point 3: How were CBC/diff chosen? Wouldn't patients have multiple CBC/diff post NCRT? Are the data gatherers blinded to the pathology results? or date of death/recurrences? Please describe in section 2.5
Response 3: Thanks for your practical comment. The blood count data were available on two time points, the first day of NCRT and the day just before OP. All the data gatherers were blinded to the pathology results and date of death or recurrences. The Material and methods (2.5) is revised as follows for explanation.
Previous version:
“Blood count data were available before and within 6-8 weeks after NCRT and before OP.”
Revised version:
“Blood count data were available on two time points, the first day of NCRT and the day just before OP. All the data gatherers were blinded to the pathology results and date of death or recurrences.”
Point 4: Did any patients receive post-surgery adjuvant chemotherapy? Wouldn't that influence DFS and OS?
Response 4: After OP, adjuvant chemotherapy has generally been recommended. In this study, all patients received adjuvant chemotherapy. The Material and methods (2.2) is revised as follows for explanation.
Previous version:
“OP was performed within 6-8 weeks after the last RT fraction.”
Revised version:
“OP was performed within 6-8 weeks after the last RT fraction. After OP, adjuvant chemotherapy has generally been recommended.”
Point 5: How do you know that the PtLR and NLR truly related to systemic inflammation/post-treatment reaction rather than bone marrow suppression from chemo-RT? Some people can develop thrombocytopenia or lymphopenia after pelvic radiation for a while
Response 5: Thanks for your great comment. The NLR and PtLR have been deemed as indicators of systemic inflammation in literature review. Few studies have focused on decreases in lymphocyte counts and their impact on response to NCRT in LARC. From these studies, we may assume that the intrinsic radiosensitivity of cancer cells and circulating lymphocytes could be correlated, which has been validated by previous studies of RT in various cancer types. The increases of NLR and PtLR after NCRT may be related to systemic inflammation, bone marrow suppression, or intrinsic radiosensitivity of cancer cells. The Discussion is revised as follows.
Previous version:
“Few studies have focused on changes in lymphocyte counts and their impact on response to NCRT in LARC. Wu et al. conducted a prospective phase III study that in-dicated that decreased lymphocyte counts during NCRT were an independent predictor of tumour regression. Heo et al. assessed the correlation between lymphocyte subpopulation counts during NCRT and tumour responses in LARC and reported that decreased lymphocyte counts during NCRT were related to good treatment responses, especially decreases in natural killer cells. Ishihara et al. verified the effects of radiation-induced apoptosis in peripheral blood lymphocytes, which was associated with the extent of tumour regression in patients with LARC after NCRT. From these studies, we may assume that the intrinsic radiosensitivity of cancer cells and circulating lymphocytes could be correlated, which has been validated by previous studies of RT in various cancer types.”
Revised version:
“Few studies have focused on decreases in lymphocyte counts and their impact on response to NCRT in LARC. Wu et al. conducted a prospective phase III study that in-dicated that decreased lymphocyte counts during NCRT were an independent predictor of tumour regression. Heo et al. assessed the correlation between lymphocyte subpopulation counts during NCRT and tumour responses in LARC and reported that decreased lymphocyte counts during NCRT were related to good treatment responses, especially decreases in natural killer cells. Ishihara et al. verified the effects of radiation-induced apoptosis in peripheral blood lymphocytes, which was associated with the extent of tumour regression in patients with LARC after NCRT. From these studies, we may assume that the intrinsic radiosensitivity of cancer cells and circulating lymphocytes could be correlated, which has been validated by previous studies of RT in various cancer types. The increases of NLR and PtLR after NCRT may be related to systemic inflammation, bone marrow suppression with decreases in lymphocyte counts, or intrinsic radiosensitivity of cancer cells.”
Point 6: DFS is difficult to interpret because disease recurrence could be local or metastatic recurrence. Could you elaborate on site of recurrence (local vs. distant) and how that could be affected from your 4 prognosis factors?
Response 6: In our database, 14 patients had local failures, and 7 of these patients suffered from distant metastases at the same follow up time. Therefore, only 7 patients had local failures only. Thirty-two patients had distant metastases. The site of recurrences were not affected from ILR, PLR, NLR, or PtLR in our analysis. The Results (3.2) is revised as follows.
Previous version:
“The median OS was not reached, and the median DFS was 36.2 months (95% confidence interval, 12.1-60.3). Distant metastases were observed in 32 (46.4%) patients.”
Revised version:
“The median OS was not reached, and the median DFS was 36.2 months (95% confidence interval, 12.1-60.3). Local failures were noted in 14 (20.3%) patients, and distant metastases were observed in 32 (46.4%) patients.”
